# Microstructural Evolution and Element Partitioning in the Phase Transformation of Ti-17 Alloy under Continuous Heating and Cooling Conditions

**Xudong An [1,2], Xin Cai [1,2], Mingpan Wan [1,2,*], Min Lei [1,2], Chaowen Huang [1,2], Fei Zhao [1,2] and Fang Huang [3]**

[1] College of Materials and metallurgy, Guizhou University, Guiyang 550025, China; anxudong@ihep.ac.cn (X.A.); 18786690021@163.com (X.C.); mlei@gzu.edu.cn (M.L.); cwhuang@gzu.edu.cn (C.H.); fzhao@gzu.edu.cn (F.Z.)

[2] Key Laboratory for Material Structure and Strength of Guizhou province, Guiyang 550025, China

[3] School of Aerospace Engineering, Guizhou Institute of Technology, Guiyang 550003, China; hfcy_215@163.com

[*] Correspondence: mpwan@gzu.edu.cn; Tel.: +86-851-83627683

**Abstract:** The microstructural evolution and alloying element partitioning in the $\alpha + \beta \leftrightarrow \beta$ phase transformation of Ti-17 alloy were explored under continuous heating and cooling conditions using the dilatometric method. Scanning electron microscopy and transmission electron microscopy were used to evaluate microstructural characteristics and trace alloying element partitioning behaviors occurring at different temperatures during heating and cooling. Results showed that the finer needle-like $\alpha$ phase first dissolved into the $\beta$ phase in the matrix with increasing temperature, while the grain boundary $\alpha$ phase first coarsened and then transformed gradually into $\beta$ phase during continuous heating. The dissolution of $\alpha$ phase of the alloy with the alloying element partitioning during continuous heating was observed. On the contrary, $\alpha_{GB}$ formed at the prior $\beta$ grain of the alloy during continuous cooling, which might be the nuclei of $\alpha$ colony, thus resulting in the formation of $\alpha$ colony in the matrix. As the temperature decreased, the elements' concentrations in the $\alpha$ and $\beta$ phases became increasingly varied due to element partition. Moreover, Al and Cr, which had higher diffusion coefficients than Mo, easily reached the concentration equilibrium of alloying elements in the $\alpha$ and $\beta$ phases, respectively. The shrinkage of dilatometric curves during heating in the Ti-17 alloy are mainly attributed to the change of $\alpha$-HCP (hexagonal close-packed) lattice to $\beta$-BCC (body-centered cubic) lattice; while the element partitioning during the $\beta \rightarrow \alpha + \beta$ transformation plays an important role in the shrinkage of the dilatometric curves of the Ti-17 alloy during cooling.

**Keywords:** Ti-17 alloy; microstructural evolution; element partitioning; heating and cooling

## 1. Introduction

Titanium and titanium alloys are ideal materials for aerospace applications, automotive engineering, and marine industries due to their properties of good crack-propagation/fatigue, excellent corrosion resistance, high hardenability, and strength-to-density ratio [1–5]. Their mechanical properties are considerably dependent on the microstructural characteristics, such as prior $\beta$ grain size and $\alpha$ content and distribution, which are highly related to thermomechanical processing conditions [6–9]. Phase transformation, an important process in heat treatment, is able to tailor the microstructural features for optimizing desirable properties [10–13]. Previous studies have investigated the $\alpha + \beta \rightarrow \beta$ and $\beta \rightarrow \alpha + \beta$ phase transformations, which are most common in many titanium alloys during heating and aging treatment [14–16]. The kinetics of phase transformation were developed and

modeled based on the Johnson–Mehl–Avrami equation in many titanium alloys to precisely predict and determine the α + β → β and β → α + β phase transformations during continuous heating and cooling, and to further reveal the mechanism of phase transformation [17,18]. In the process of α + β ↔ β transformation, the crystalline structures and chemical compositions of the phases are altered, resulting in changes in the physical properties of the alloy, which could be used to investigate the phase transformation mechanism, such as thermal analysis, the dilatometric method, and electrical resistivity [19–22]. In the past, thermodilatometry was not considered as a very sensitive method due to the low volume variation associated with phase transformation in the titanium alloy. However, with the development of high-resolution dilatometers, this method has been widely used to study phase transformation in titanium alloy. Slight volume contraction resulting from phase transformation during heating and cooling has been found.

In the current work, the microstructure evolution and element partitioning in the α + β ↔ β phase transformation of Ti-17 alloy was investigated under continuous heating and cooling conditions using the dilatometric method. Scanning electron microscopy (SEM) and transmission electron microscopy (TEM) were used to evaluate the microstructural characteristics and alloying element partitioning behavior occurring at different temperatures. The experimental data can be used to enhance the understanding of phase transformation.

## 2. Experimental Procedures

The Ti-17 alloy bar, provided by Guizhou Anda Aviation Forging Co., Ltd., P.R. China, has a nominal composition of Ti-5Al-2Sn-2Zr-4Mo-4Cr. The β transus temperature of the alloy was determined by dilatometric method using a DIL-805 A/D dilatometer (Bähr, Germany) at approximately 1173 ± 5 K. The microstructure of the as-received material was a typical basket-weave characteristic and consisted of coarse α platelets and fine needle-like α phase distributed on the β matrix, as shown in Figure 1.

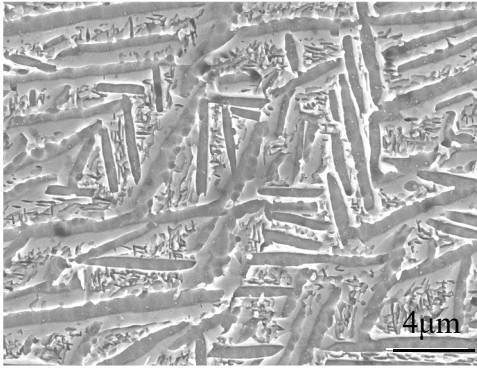

**Figure 1.** Microstructure of the as-received Ti-17 alloy.

The cylindrical specimens with 4-mm diameter and 10-mm length for the dilatometric study were obtained from the alloy bar. A Bähr DIL-805 A/D dilatometer with inductive heating device was used to measure the thermal dilation of the Ti-17 alloy during continuous heating and cooling. The temperature was controlled within ±0.05 K. The specimens were heated up to 1373 K for 15 min under vacuum conditions at a heating rate of 0.2 K/s and then cooled to room temperature at a cooling rate of 0.1 K/s. In order to observe the microstructure evolution and element partition, we heated up the chosen specimens up to 933, 998, 1058, 1118, and 1178 K at a heating rate of 0.2 K/s and then cooled rapidly using argon gas (99.999%) to freeze the microstructure and composition. The two specimens were heated up to 1373 K for 15 min at a heating rate of 0.2 K/s, subsequently cooled to 1003 K and 893 K, respectively, at a cooling rate of 0.1 K/s; and then cooled rapidly using the above-mentioned method. The metallographic specimens were prepared after mechanical grinding and polishing and etched with Kroll's reagent (10 vol% HF, 20 vol% HNO₃, and 50 vol% H₂O). The resultant microstructural characterization was carried out using a SUPRA40 scanning electron microscope (SUPRA40, Carl

Zeiss AG, Oberkochen, Germany). The Image-Pro Plus 4.5 was used to analyze the volume fraction of alpha phase in microstructure. The sliced metallographic disks (0.5-mm thick) were prepared and mechanically thinned to a thickness of about 80 μm. After ion polishing, a Titan Cubed Themis G2 300 (special aberration corrected transmission electron microscope, ACTEM, FEI, USA) at 200 kV was used to observe the microstructure and analyze the chemical composition.

## 3. Results and Discussion

### 3.1. Dilatometric Analysis

Figure 2a shows the dilatometric curves and corresponding derivative curve of the Ti-17 alloy heated from room temperature to 1373 K at a heating rate of 0.2 K/s. The dilatometric curve could be divided into three sections in terms of its features. The Ti-17 alloy dilated linearly from room temperature to 933 K with increasing heating temperature. This phenomenon can be attributed to the thermal expansion of the crystal lattice in the Ti-17 alloy during continuous heating. When the temperature was above 933 K, the Ti-17 alloy started to shrink with the decrease in the slope of dilatometric curve, which could be result from $\alpha + \beta \rightarrow \beta$ phase transformation in the alloy. After $\alpha + \beta \rightarrow \beta$ phase transformation finished, the length of specimen continues to dilate as the temperature increases. Figure 2b shows the dilatometric curves and corresponding derivative curve of the Ti-17 alloy cooled from 1373 K to room temperature at a heating rate of 0.1 K/s. The Ti-17 alloy first shrank linearly, then rapidly shrank at around 993 K, and finally ended below approximately 873 K. This phenomenon is in agreement with that of Ti-1300 alloy during continuous cooling [16]. The shrinkage phenomenon was found on the dilatometric curves of Ti-17 alloy during continuous heating and cooling. It is well-known that the change in length of titanium alloy is mainly dependent on many factors, including phase transformation, thermal expansion, element partition, and defect annihilation. Previous research reported that this shrinkage might be associated with the $\alpha + \beta \rightarrow \beta$ or $\beta \rightarrow \alpha + \beta$ phase transformation and element partition in titanium alloy [16,23].

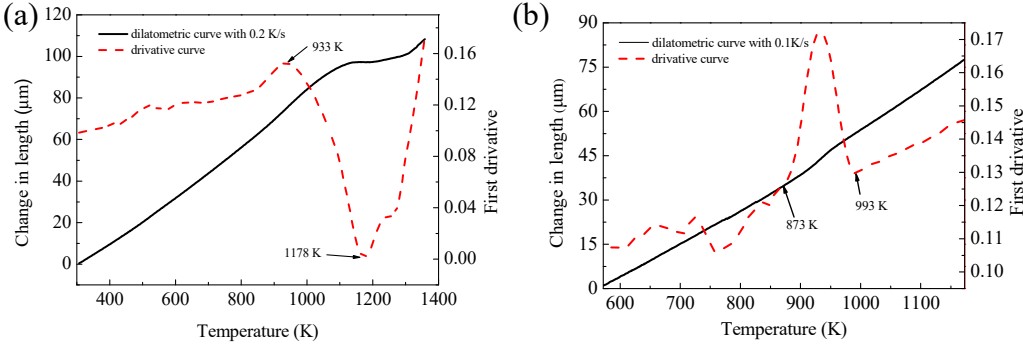

**Figure 2.** Dilatometric curves and corresponding derivative curves of Ti-17 alloy during heating (**a**) and cooling (**b**).

To explain the dilatometric behaviors of the alloy during heating and cooling, the chosen specimens were heated to different temperatures corresponding the contraction of dilatometric curves of the alloy, followed by rapid cooling for microstructural observation and composition mapping.

### 3.2. Microstructural Evolution

Figure 3 shows the microstructure of the Ti-17 alloy heated to 998, 1058, 1118, and 1178 K at a heating rate of 0.2 K/s and then rapidly cooled to room temperature at a cooling rate of 80 K/s. A number of α platelets uniformly existed in the matrix of the alloy heated up to 998 K, and the fine needle-like α phase in the matrix transformed into β phase. The volume fraction of the remaining α platelets with mean thickness of approximately 0.8 μm was 42.8%, as shown in Figure 3a. With the increase in temperature, the thicker plate-like α phases started to dissolve into the matrix and the

volume fraction of α phase further decreased in the matrix. However, a small amount of α platelets that accumulated at prior β grain boundaries of the alloy coarsened when heated up to the higher temperature. These phenomena can be attributed to the volume fraction of α phase in the alloy decreasing on the basis of the level law and the diffusion coefficients of atoms increasing with the increase in temperature. Furthermore, the atoms preferentially segregated along the grain boundaries because of their higher energy state, resulting in $\alpha_{GB}$ (α phase on grain boundaries) coarsening, as shown in Figure 3b,c. When the alloy was heated up to 1178 K and then rapidly cooled to room temperature at a cooling rate of 80 K/s, the microstructure almost consisted of a large amount of β phase and only 3.4% α phase, as shown in Figure 3d. The above-mentioned microstructural evolution produced in the temperature range corresponding the shrinking of the dilatometric curve of the alloy during heating. Two representative bright-field TEM images of the alloy heated to 933 and 1178 K, along with selected area diffraction patterns, are shown in Figure 3e and f, respectively. In addition to the observed plate-like α phase distributed on the matrix, a large number of dislocations also piled up in the near area α/β interfaces in the alloy, as shown in Figure 3e. The dislocations disappeared when the heating temperature was increased due to recovery.

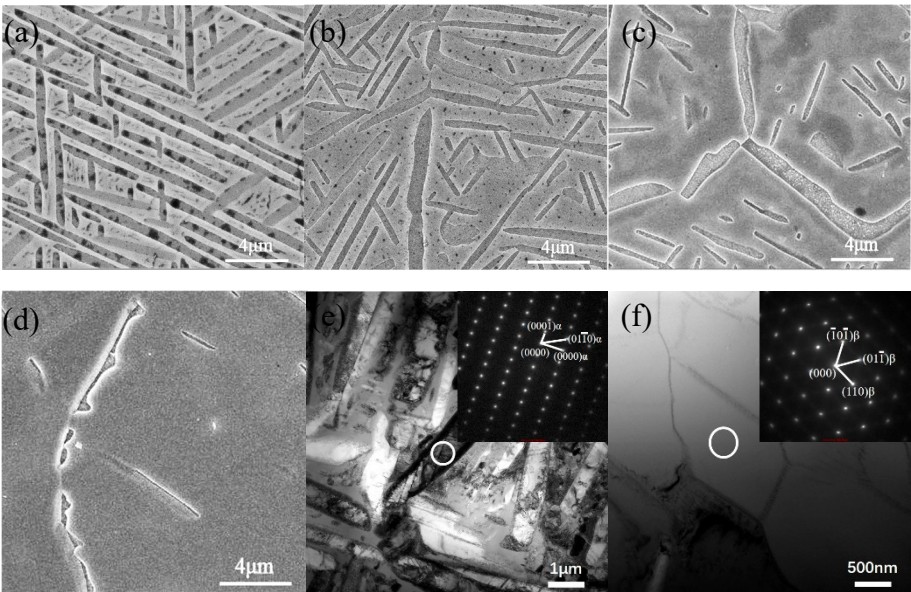

**Figure 3.** Microstructure of Ti-17 alloy heated to different temperature at a heating rate of 0.2 K/s: (**a**) 998 K; (**b**) 1058 K; (**c**) 1118 K; (**d**) 1178 K; (**e**) microstructure of Ti-17 alloy heated to 933 K and corresponding diffraction pattern and (**f**) microstructure of Ti-17 alloy heated to 1178 K and corresponding diffraction pattern.

Figure 4 shows the microstructure of the Ti-17 alloy cooled to 1003 and 893 K, respectively, at a cooling rate of 0.1 K/s and then rapidly cooled to room temperature at a cooling rate of 80 K/s. The α phase started to precipitate along the prior β grain boundaries in the alloy, as indicated by the arrow in Figure 4a. This $\alpha_{GB}$ could be the nuclei of the α phase growing side by side toward the prior β grain interior, finally resulting in the formation of α colonies in the alloy, as shown in Figure 4a. The microstructure of the α colony tip observed by TEM is illustrated in Figure 4b. The α precipitates with particles formed directly in the phase in the alloy cooled to 1003 K. When the alloy was cooled to 893 K, the microstructure in the prior β grain mainly consisted of α platelets with different lengths and widths, as shown in Figure 4c, indicating that the fraction volume of α phase gradually increased as the temperature decreased. The α platelets were regularly arranged on the matrix, and the angle between α platelets was approximately 60°, as shown in Figure 4d. The formation and growth of the α phase was also produced in the temperature range corresponding the shrinking of the dilatometric curve of the alloy during cooling.

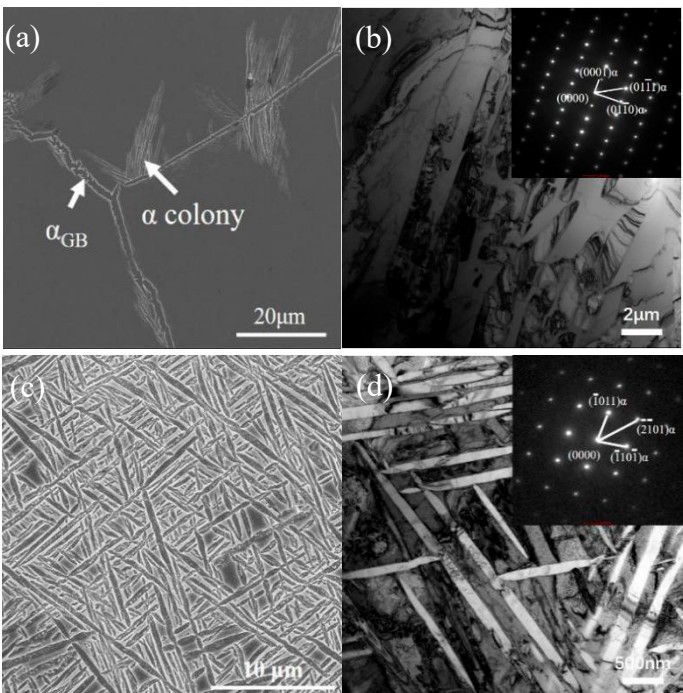

**Figure 4.** Microstructure of Ti-17 alloy cooled to 1003 K and 893 K at a cooling rate of 0.1 K/s: (**a**) SEM micrograph of Ti-17 alloy cooled to 1003 K; (**b**) microstructure of the α colony tip in the (a) and corresponding diffraction pattern; (**c**) SEM micrograph of Ti-17 alloy cooled to 893 K; (**d**) microstructure of the alloy cooled to 893 K and corresponding diffraction pattern.

As above-mentioned, the $\alpha + \beta \rightarrow \beta$ transformation occurred during the range of temperature from 933 to 1178 K during continuous heating for the alloy, while the $\beta \rightarrow \alpha + \beta$ transformation occurred during the range of temperature from 993 to 873 K during cooling. Additionally, the contraction effect of the dilatometric curve of the alloy just occurred during the temperature range. It can be confirmed that the shrinkage phenomenon must be associated with $\alpha + \beta \rightarrow \beta$ or $\beta \rightarrow \alpha + \beta$ transformation. It is well-known that pure titanium exhibits an allotropic phase transformation, changing from a hexagonal close-packed crystal structure (α phase) to a body-centered cubic crystal structure (β phase) at higher temperatures. The lattice parameter is about $a_\beta = 3.26$ Å, so the lattice volume is $V_\beta = 34.64$ Å$^3$ for two atoms in the unit cell. The mean values are $a_\alpha = 2.93$ Å and $c_\alpha = 2.4.48$ Å, so the lattice volume is $V_\alpha = 34.79$ Å$^3$ for two atoms in the unit cell [24]. Thus, the contraction effect of the dilatometric curve of the Ti-17 alloy during heating resulted from the change of α-HCP lattice to β-BCC lattice when the $\alpha + \beta \rightarrow \beta$ transformation occurred. However, the polymorphic transformation could not be simply used to explain the contraction response of the dilatometric curve of the alloy during cooling.

### 3.3. Local Composition Analysis

Figure 5 shows the local compositions of the α and β phase of the Ti-17 alloy heated to 933 K at a heating rate of 0.2 K/s and then rapidly cooled to room temperature at a cooling rate of 80 K/s to freeze the chemical composition and microstructure. It is well-known that the α and β phase contain different contents of alloying elements in the alloy. The distributing characteristics of each alloying element varied between the α and β phase based on the intrinsic type of alloying element, as shown in Figure 5b–f. Figure 5g shows that the α phase contained more α-stabilizer Al element than the β phase, whereas the β phase in the alloy had more concentration of β-stabilizers like Cr and Mo elements than the α phase. Sn and Zr, as neutral elements, were homogeneously distributed in the α and β phase. The maximal concentration of Al in the interior of α phase reached approximately 12.7 at. pct, which was higher than that in the nominal composition of the Ti-17 alloy, whereas the Cr and Mo contents

were around 0.44 and 0.35 at. pct, respectively. On the contrary, the β phase contained 5.72 at. pct Al element, which is lower than the average content of the alloy. The β phase also contained 9.98 at. pct Cr and 6.83 at. pct Mo, which were higher than those average contents of the alloy. The results indicated alloying elements partitioning produced among the α and β phase when the alloy was heated above a certain temperature. α-stabilizer Al element in the α phase diffused into the β phase due to the α + β → β transformation during heating, leading to the increase in composition concentration of α-stabilizer in the β phase.

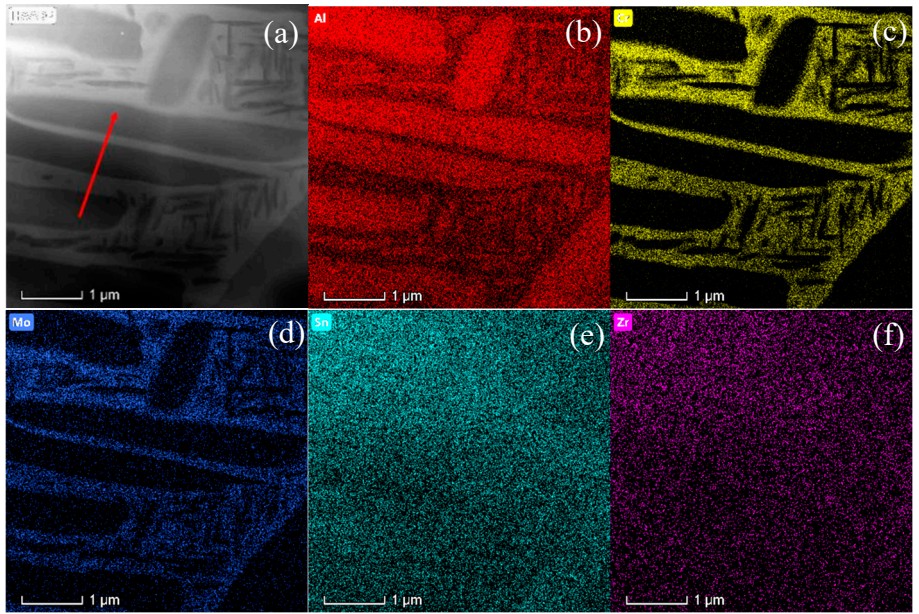

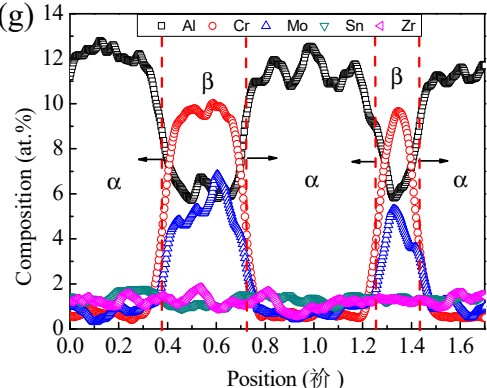

**Figure 5.** Microstructure and local composition mapping in the sample heated to 933 K: (**a**) HAADF (high-angle annular dark field image), (**b**) Al, (**c**) Cr, (**d**) Mo, (**e**) Sn, (**f**) Zr, and (**g**) composition profiles along the red line in (**a**).

Figure 6a shows TEM bright-field images the frozen microstructure and local compositions of the Ti-17 alloy after heating at a heating rate of 0.2 K/s up to 1178 K and then rapidly cooled to room temperature at a cooling rate of 80 K/s. It can be seen that a pattern of parallel domains existed in the β matrix of the alloy. A nominal period of 27 nm was obtained by considering the distance between the peak intensities of the domains. As mentioned previously, when the alloy was heated to 1178 K, the microstructure of the alloy primary consisted of the β phase. Thus, it could be considered that the periodic domains were transformed from the β phase during rapidly cooling for the alloy, which are associated with the composition modulations of spinodal decomposition of the β phase [25]. Similar microstructures stemming from the spinodal decomposition mechanism for β phase were obtained

upon quenching Cu-Ti alloys [26,27]. Figure 6b showed domains of different compositions, namely β′ and β that are poor and rich in β stabilizers, respectively. Moreover, the interface between two phases is not sharp but very diffuse, there is still an effective interfacial energy contribution. The magnitude of this energy depends on the composition gradient across the interface, and for this reason it is known as a gradient energy [28].

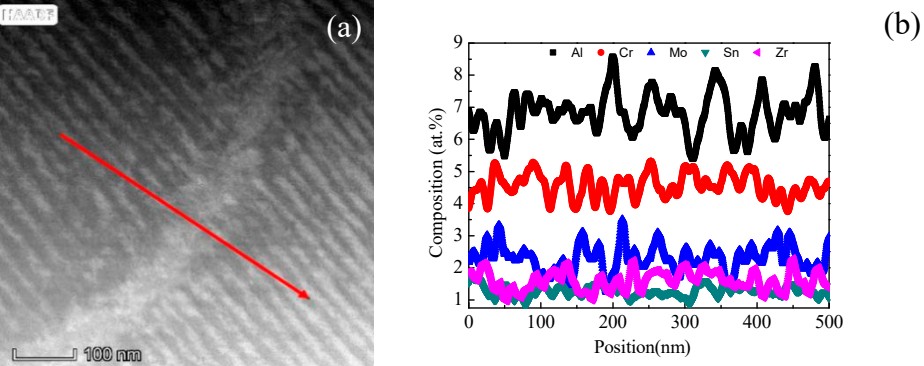

**Figure 6.** Microstructure in the sample heated to 1178 K: (**a**) HAADF and (**b**) composition profiles along the red line in (**a**).

Figure 7 shows the local composition in the α and β phases of the Ti-17 alloy cooled to 1003 K at a cooling rate of 0.1 K/s and then rapidly cooled to room temperature at a cooling rate of 80 K/s. The microstructure was selected from the near tip of the α colony, indicating that the α platelets were found under this condition. The Al content in the α phase, Cr, and Mo contents in the β phase slightly decreased. Therefore, the difference in the element concentration of α and β phases was lower than that at 933 K during heating. The major observable difference was that the local compositions in α and β phase interiors displayed more noticeable fluctuations, as shown in Figure 7g, which is attributed to the absorbed Al element from the surrounding metastable β phase during the formation and growth of α precipitations. Furthermore, during the process, the β-stabilizers Cr and Mo were spat out from the α phase and diffused into the β phase as a consequence of free energy. However, the diffusion abilities of Al, Cr, and Mo in the α and β phases varied due to the crystalline structure of both phases and the type of alloying elements. Previous research reported that the diffusion coefficient of Al in the β phase at moderate temperature ($\sim 10^{-13}$ m$^2$ s$^{-1}$ at 1173 K) is four orders of magnitude faster than the that in the α phase ($\sim 10^{-17}$ m$^2$ s$^{-1}$ at 1173 K) [29]. In addition, the diffusion coefficient of Al ($\sim 4.5$–$7.43 \times 10^{-13}$ m$^2$ s$^{-1}$ at 1473 K) in the β phase is at least three-times faster than that of Mo ($1.45 \times 10^{-13}$ m$^2$ s$^{-1}$ at 1473 K). However, the mean diffusion coefficient of Cr ($1.56 \times 10^{-12}$ m$^2$ s$^{-1}$ at 1473 K) is approximately two-times larger than that of Al [18]. In the present study, the diffusion coefficient of Mo in the β phase was the lowest among the three elements. The concentrations of Al in the α phase and Cr and Mo in the β phase were considered as concentration in equilibrium when the alloy was heated to 933 K. Nonetheless, the Cr concentration in the β phase rapidly reached the concentration in equilibrium when cooled to 1003 K, whereas the maximum concentration of Mo in the β phase was approximately 4.23 at. pct.

Figure 8 shows the local compositions in the α and β phases of the Ti-17 alloy cooled to 893 K at 0.1 K/s and then rapidly cooled to room temperature at 80 K/s. Compared with 1003 K, the volume fraction of the α phase further increased and the α phase distributed regularly on the matrix. It is found that the distribution characteristic of alloying element in the α and β phases was more obvious. The α/β interface continued to move from the α to β phase in the matrix. These phenomena were attributed to the redistribution of α-stabilizer and β-stabilizer during slow cooling. Moreover, the contents of Al in the α phase and Cr and Mo in the β phase further increased in Figure 8g, which is basically closed to the concentration in equilibrium.

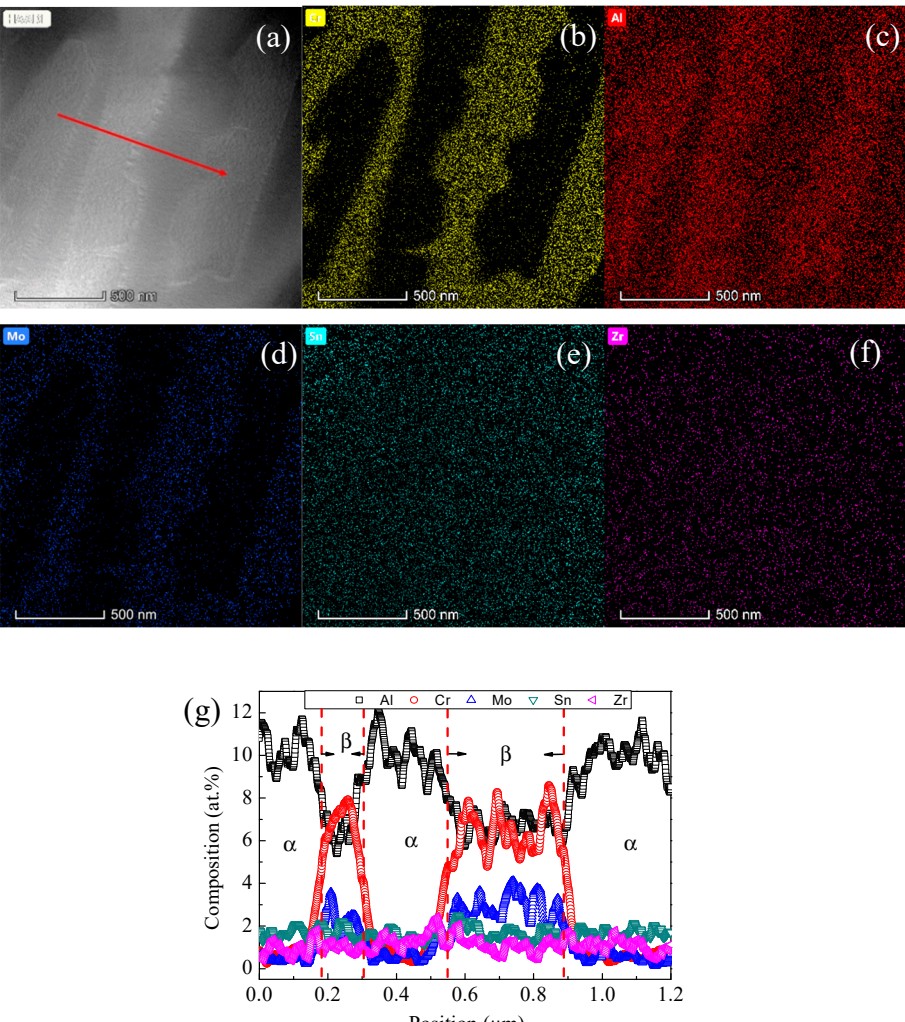

**Figure 7.** Microstructure and local composition mapping in the sample heated to 1003 K: (**a**) HAADF, (**b**) Al, (**c**) Cr, (**d**) Mo, (**e**) Sn, (**f**) Zr, and (**g**) composition profiles along the red line in (**a**).

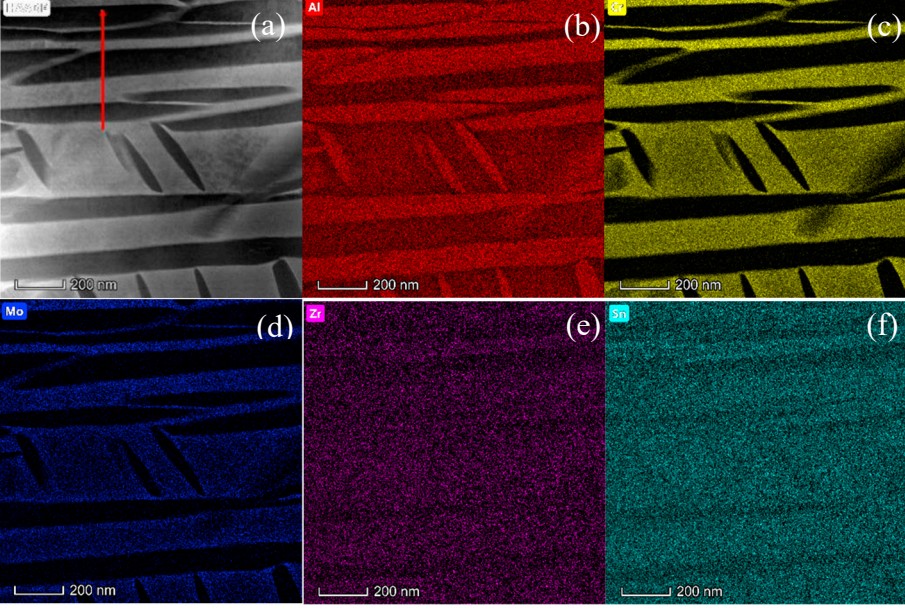

**Figure 8.** *Cont.*

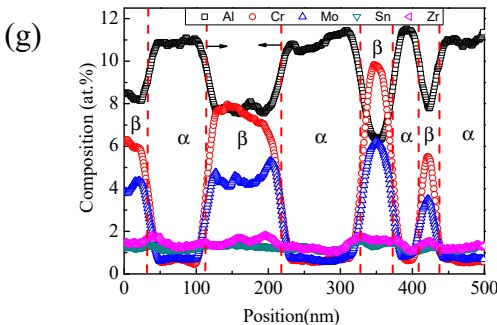

**Figure 8.** Microstructure and local composition mapping in the sample cooled to 893 K: (**a**) HAADF, (**b**) Al, (**c**) Cr, (**d**) Mo, (**e**) Sn, (**f**) Zr, and (**g**) composition profiles along the red line in (**a**).

In fact, the element partitioning during the α + β → β or β → α + β transformation results in the volume change of the titanium alloy. α-stabilizer Al element in the α phase diffused into the β phase due to the α + β → β transformation when the alloy was heated over a certain temperature, leading to the increase in composition concentration of α-stabilizer in the β phase and the decreased in composition concentration of β-stabilizer in the β phase. Whereas the composition concentration of α-stabilizer in the β phase decreased and the composition concentration of β-stabilizer in the β phase increased during β → α + β transformation with decreasing temperature. The Al atoms with the larger atomic radius tend to concentrate in α phase, while the Mo, Cr, and V with the smaller atomic radii prefer to concentrate in the β phase [23]. During the β → α + β transformation, the β phase enrichment in β-stabilizers, particularly Mo and V, leads to a decrease of the β phase lattice parameter [24]. And the resulting shrinkage is of much greater amplitude than the dilatation produced by the change of β-BCC lattice to α-HCP lattice. When the Ti-17 alloy was cooled at 0.1 K/s, a contraction effect of high amplitude was observed on the dilatometric curve. Consequently, the element partitioning during the β → α + β transformation plays an important role in the shrinkage of the dilatometric curves of the Ti-17 alloy during continuous cooling.

## 4. Conclusion

- A finer needle-like α phase easily dissolves into β phase in the matrix with increasing temperature, whereas the $\alpha_{GB}$ first coarsens and then transforms gradually into β phase during continuous heating.
- The $\alpha_{GB}$ first form at the prior β grain of the alloy during continuous cooling, which might be the nuclei of α colony, resulting in the formation of α colony in the matrix. As the temperature decreases, the concentrations in the α and β phases become increasingly varied due to element partitioning.
- The shrinkage of dilatometric curves during heating in the Ti-17 alloy are mainly attributed to the change of α-HCP lattice to β-BCC lattice during α + β ↔ β phase transformation; while the element partitioning during the β → α + β transformation plays an important role in the shrinkage of the dilatometric curves of the Ti-17 alloy during continuous cooling.

**Author Contributions:** Conceptualization, M.W.; methodology, M.L.; validation, F.Z. and C.H.; formal analysis, X.C.; investigation, X.C.; data curation, X.C.; writing—original-draft preparation, X.C. and X.A.; writing—review and editing, M.W.; visualization, C.H. and M.L.; project administration, F.Z.; funding acquisition, F.Z. and F.H. All authors have read and agreed to the published version of the manuscript.

**Funding:** This project was financially supported by the Nation Natural Science Foundation of China (Grant No. 51401058 and 51801037) and the Natural science foundation of Guizhou province (Grant No. [2017]1023) and the Project of Science and Technology of Guizhou province (Grant No. [2017]2313). We also appreciate the Talent Introduction Project of Guizhou Institute of Technology (Grant No. XJGC20190950).

**Conflicts of Interest:** The authors declare no conflict of interest.

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
