# Peer review of "Microstructural Evolution and Element Partitioning in the Phase Transformation of Ti-17 Alloy under Continuous Heating and Cooling Conditions"

_metals, doi:10.3390/met10081054_

Round 1
Reviewer 1 Report
This is a nicely written paper focusing on the microstructural changes of the Ti-17 alloy that occur as the temperature increase and decrease. The experiment was simple but seemed sufficient. The conclusion was also simple supported but the clear images, but there is a concern on the description regarding the relationship between phase transformation and shrinkage. In line 144 – 146, authors stated that “The formation and growth of α phase also produced in the temperature range corresponding the shrinking of the dilatometric curve of the alloy during cooling.” That seemed correct. In the abstract and conclusions, authors also stated that “The shrinkage of the dilatometric curves during heating and cooling was attributed to α↔α+β phase transformation”. That possibly correct, but that would be interpretation of the similarity in those temperature ranges. The phase transformation would be one possible explanation of the dilatometric curve.
Other minor issues:
Please spell out αGB when it first appears, though it would be obvious.
Figure 3
The signs of (a)-(f) were not put in the appropriate places.
Figure caption for Fig 3 said Ti-75, but I guess that was Ti-17.
Figure 4
The signs of (a)-(d) were not put in the appropriate places.
Figure 6
The signs of (b) were not put in the appropriate places.
Figure 8
The signs of (a)-(c) were not put in the appropriate places.
Figure caption said that the sample heated to, but it might be ”cooled to”.
Reviewer 2 Report
Line 88: You state that dilatometric curve can be divided in three section. Please determine and explain individual section. The label of individual section and α↔β transformation in dilatometric curves (Fig 2a,b) can be helpful for readers.
Line 110: How did you measure the volume fraction of α platelets? It should be describe in Experimental part.
Did you try to investigate influence of alloying elements partitioning on hardness? What do you mean about it?
Reviewer 3 Report
The paper was prepared not very carefully. There is a lot of "problems" with figures and some "unclears":
- Fig. 1. axis description and legend equal revision
- Fig. 3 unvisible literal marks on fig. 3
- Fig. 4. In 151 line (fig. description: d instead of b) and problem with literal marks (,a,b, c, d) on fig.4
- Fig. 8 a,b,c, not visible; (au.% or at.% like in fig. 7 and 5)
- l.206-there is no position [24] in references
- please check format of authors in [22] references position
- lack of methodology for determining the volume fraction of alpha phase as well as diffusion coefficient ("In the present study, the diffusion coefficient of Mo in the β phase was the....")
- sentences: l. 156-7 and 159-161 are not a results...its obvious.
- l. 178 "Fig. 6(a) shows the periodic domains ...[19]." -It is a result or comment (references)? (see conclusion 2)
- "In the current work, [...] the α↔α+β phase transformation of Ti-17 alloy was investigated"-results presented in part: l.134-152 seems to concern with β↔α+β transformation.
- spell check required
After corrections, I accept this article for publication in Metals
